# Immunophenotype and antitumor activity of cytokine-induced killer cells from patients with hepatocellular carcinoma

Chan-Keng Yang[1,2], Chien-Hao Huang[2,3], Ching-Hsun Hu[1], Jian-He Fang[3], Tse-Ching Chen[4,5], Yung-Chang Lin[1,5]*, Chun-Yen Lin[3,5]

1 Division of Hematology-Oncology, Department of Internal Medicine, Linkou Medical Center, Chang Gung Memorial Hospital, Kweishan, Taoyuan, Taiwan, 2 Graduate Institute of Clinical Medical Sciences, College of Medicine, Chang Gung University, Kweishan, Taoyuan, Taiwan, 3 Division of Gastroenterology-Hepatology, Department of Internal Medicine, Linkou Medical Center, Chang Gung Memorial Hospital, Kweishan, Taoyuan, Taiwan, 4 Department of Pathology, Linkou Medical Center, Chang Gung Memorial Hospital, Kweishan, Taoyuan, Taiwan, 5 College of Medicine, Chang Gung University, Kweishan, Taoyuan, Taiwan

* yclinof@cgmh.org.tw

**Data Availability Statement:** All relevant data are within the paper.

**Funding:** This work was financially supported by CMRPG3F1301, CMRPG3J0871, CMRPG5J0101

## Abstract

### Background

Cytokine-induced killer (CIK) cells are heterogeneous lymphocytes from human peripheral blood mononucleated cells (PBMCs) co-cultured with several cytokines. The main purpose of this study is to evaluate the functional characteristics and anticancer ability of CIK cells from hepatocarcinoma (HCC) patients.

### Methods

CIK cells were activated ex-vivo and expanded from PBMCs from HCC patients. The immunophenotype and the ex-vivo killing ability of CIK cells were evaluated. Human CIK cells were intravenously injected into NOD/SCID mice to evaluate the in vivo anticancer ability.

### Results

More than 70% of CIK cells were CD3+CD8+, and 15%–30% were CD3+CD56+. These cells expressed an increased number of activated natural killer (NK) receptors, such as DNAM1 and NKG2D, and expressed low-immune checkpoint molecules, including PD-1, CTLA-4, and LAG-3. Among the chemokine receptors expressed by CIKs, CXCR3 and CD62L were elevated in CD8+ T cells, representing the trafficking ability to inflamed tumor sites. CIK cells possess the ex-vivo anticancer activity to different cell lines. To demonstrate in vivo antitumor ability, human CIK cells could significantly suppress the tumor of J7 bearing NOD/SCID mice. Furthermore, human immune cells could be detected in the peripheral blood and on the tumors after CIK injection.

### Conclusions

This study revealed that CIK cells from HCC patients possess cytotoxic properties, and express increased levels of effector NK receptors and chemokine molecules and lower

and CMRPG5K0221 from Medical Research Project Fund, Chang Gung Memorial Hospital, Taiwan and MOST 107-2314-B-182A-157 from Ministry of Science and Technology, Taiwan. All the funding of support received during this study. There was no additional external funding received for this study. The funders had no role in study design, data collection and analysis, decision to publish, or preparation of the manuscript.

**Competing interests:** The authors have declared that no competing interests exist.

**Abbreviations:** CIK, cytokine-induced killer; PBMCs, peripheral blood mono-nucleated cells; HCC, hepatocellular carcinoma; HBV, hepatitis B virus; HCV, hepatitis C virus; PD-1, Programmed cell death-1; IFN-γ, interferon-γ; IL-2, interleukin-2; MHC, major histocompatibility complex; RPMI, Roswell Park Memorial Institute; PBS, phosphate-buffered saline; PMA, phorbol 12-myristate 13-acetate; LDH, lactate dehydrogenase; FBS, fetal bovine serum; SPF, S-phase fraction; HBSS, Hanks' balanced salt solution; ATP, adenosine triphosphate; IQR, interquartile range; GMP, good manufacture product; OS, overall survival; DFS, disease-free survival.

levels of suppressive checkpoint receptors. CIK cells can suppress human HCC ex-vivo and in vivo. Future clinical trials of human CIK cell therapy for HCC are warranted.

## Introduction

Hepatocellular carcinoma (HCC) is the sixth most common cancer and the third leading cause of cancer-related death with a particularly high prevalence in Asia. Surgical resection and other local therapeutic strategies are considered curative therapies for localized disease, however, more than half of these would recur even with this type of curative treatment [1]. The results of other treatments for palliative purpose remain a huge unmet need despite the recent success of targeted and immunotherapy in a small fraction of patients [2]. HCC patients are often found to have functional deficiencies in host adaptive and innate immunity responses [3]. Most of the patients affected by HCC in Taiwan are associated with viral infections, such as the HBV or HCV. Evidence from PD-1 blockade therapy has suggested that viral-associated HCC is also associated with better responses [4, 5]. This implies that the immune systems of these patients may harbor certain antitumor responses [5, 6]. Adoptive immunotherapy that expands and rejuvenate autologous immune cells may be potentially used as an efficient adjuvant anticancer therapy for patients with HCC [7].

Cytokine-induced killer (CIK) cells are a heterogeneous subset of ex-vivo expanded T lymphocytes which express both the T-cell marker CD3 and NK cell receptors [8]. Patient peripheral blood mononuclear cells (PBMCs) are stimulated and cultured with a cytokine cocktail comprising interferon-γ (IFN-γ), interleukin-2 (IL-2), anti-CD3, and/or IL-1 for 2–4 weeks. These cells are able to expand up to 1000-fold. This expansion allows an easy adoptive transfer of large amounts of cells compared with conventional adoptive cellular therapy [8, 9]. The final products of these cells are usually CD3+CD8+ T cells, among which 15%–30% are also CD56+[10]. Interestingly, most of these cells will express NK cell surface receptors, such as NKG2D and DNAM-1 [11–13]. Prior studies have confirmed that CIK cells may kill tumor cells according to a nonmajor histocompatibility-complex (MHC)-restricted mechanism, thus suggesting the importance of NK surface receptors in mediating the killing process [11, 13–15].

Preclinical data and initial clinical studies support CIK cells as a promising cell population for adoptive immunotherapy [16–18]. The clinical efficacy has a broad coverage, especially on hematogenous malignancy [19, 20], melanoma [21], lung [16], and HCC, which are among the most reported types of cancers [12, 22]. A few clinical trials have shown that CIK cell therapy could prolong survivals in HCC in metastatic or adjuvant settings [23, 24]. It is interesting that treatments with CIK cells have also been associated with decreases in HBV viral loads [25]. Recently, a Korean, randomized, phase-3 clinical trial showed adjuvant adoptive CIK cell therapy for patients who underwent curative treatment for HCC could increase recurrence-free and overall survival [24]. A renowned worldwide cancer treatment guideline has adopted this result, and this CIK therapy has become a recommended treatment for HCC adjuvant therapy [26].

The clinical benefit of CIK therapy has been documented, however, there is uncertainty about common applications of this strategy to routine practice. Key issues for clinical application of adoptive immunotherapy include the identification of methodologies to scale-up the expansion process to meet quantity and quality requirements for immunotherapy; the expressed markers that could represent non-MHC restricted cytotoxicity include the CD56

molecule, other candidate markers and the ability for trafficking into tumor sites. In this study, we evaluated the immunophenotype and ex-vivo and in vivo killing ability of CIK cells based on a modified protocol established in our laboratory. This work sets up a reference laboratory standard and can be applied as a quality index for future clinical level manufacturing.

## Materials and methods

### Study subjects

We collected PBMCs samples form adult HCC patients for the expansion of CIK cells. All of the HCCs were diagnosed by cytological or pathological evaluations, or according to the criteria of the American Association for the Study of Liver Disease [27]. We excluded patients with severe alcoholic hepatitis or acute or chronic liver failure, non-HBV, HCV, or alcohol-related HCC, and patients with concurrent evidence of sepsis.

### Ethics statement

This study was approved by the Institutional Review Board of Chang Gung Medical Foundation. Approval from the institutional review board was obtained for the analysis of this series (IRB: 201600271B0). All participants provided their written informed consent to participate in this study. All animal procedures followed the *Guide for the Care and Use of Laboratory Animals* as promulgated by the Institute of Laboratory Animal Resources, National Research Council, National Academy of Science (United States), and were approved by the Animal Care and Use Committee of the Chang Gung Memorial Hospital at Linkou Medical Center. In all mice, tissue collection procedures were initiated after mice had either been euthanized or were under deep anesthesia and unresponsive to all stimuli, and all precautions were taken to minimize suffering. Mice were placed into a chamber filled with vapor of the anesthetic isoflurane until respiration ceased followed by 100% carbon dioxide into a bedding-free cage initially containing room air with the lid closed at a rate sufficient to induce rapid anesthesia, with death occurring within 2.5 minutes.

### Isolation and culture of CIK cells

PBMCs were isolated from HCC patients based on the Ficoll–Paque plus density gradient. The cells were washed twice with RPMI-1640. PBMCs were cultured with RPMI-1640+10%FBS medium with anti-CD3 antibody (50 ng/ml), IL-1α (100 U/ml), and IFN-γ (1,000 U/ml) at 37˚C with 5% $CO_2$ for 24 h. From days 2 and 3, IL-2 (300 U/ml) was added to the medium. We then changed the IFN-γ and IL-2 containing medium every 5 days. On day 14, CIKs were harvested (viability: 90%), and the phenotypes and killing abilities of CIK cells were evaluated. The preparation of CIK cells is illustrated in Fig 1.

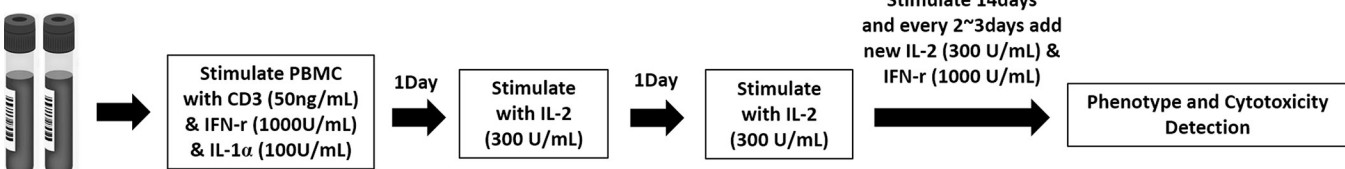

**Fig 1. Flow chart showing the preparation of cytokine-induced killer (CIK) cells.** After harvesting peripheral blood mononucleated cells (PBMCs), cells were stimulated with Anti-CD3 Ab, interleukin (IL)-1α, and interferon (IFN)-γ on day1. On days 2 and 3, cells were stimulated with IL2, and IFN-γ and IL-2 every 2–3 days until day 14. The final products were used for phenotypic analysis and cytotoxicity assays.

## Phenotypic analysis

We resuspended $5 \times 10^5$ CIK cells in 20 μl of phosphate-buffered saline (PBS) mixed with 2% newborn calf serum and 1% sodium azide. The following antibodies were used for flow cytometric staining: conjugated anti-CD4-Pe-cy7, anti-CD3 Per CP conjugated anti-CD8-FITC, anti-CD25-PE, anti-CD56-APC, anti-CD69-PE, anti-CD103-PE or-FITC, anti-CTLA4-PE, biotin-conjugated all purchased from BD Pharmingen. Anti-perforin-FITC from BD; anti-granzyme B-FITC, anti-lag3-PE, PE- or allo-phycocyanin-conjugated anti-Foxp3, isotype control human IgG2a, and human IgG2b from eBioscience. Single-cell suspensions were incubated with Abs per the manufacturer's instructions. All samples were acquired using a FACSCalibur flow cytometer (BD Bioscience) and analyzed with the software FlowJo (Tree Star Inc., Ashland, OR, USA). For intracellular staining, PBMCs were restimulated with a 50 ng/ml cocktail of phorbol 12-myristate 13-acetate (PMA) and 500 ng/ml ionomycin mix in RPMI1640 with 10% fetal bovine serum for 5 h. In total, 10 μg/ml brefeldin A (Sigma–Aldrich) was then added during the last 2 h. Cells were initially stained for surface markers by discarding supernatant and resuspending cells in 200 μl of Cytofix/Cytoperm™ (BD Biosciences) solution for 20 min at 4°C. Cells were washed twice in Perm/Wash buffer (BD Biosciences). Resuspended cells in 100 μl of Perm/Wash buffer could then be stored at 4°C in the dark. Appropriately conjugated fluorescent antibodies were added at predetermined optimum concentrations.

## Cytotoxicity assessment

The cytotoxicity of CIK cells was tested from the HCC patients. Cell lines K562, HepG2, and J7 from the American Type Culture Collection were selected as target cell lines. For K562 and HepG2 cytotoxicity assessments, we used flow cytometry to determine the expression Annexin V and 7-AAD with different effector-to-target ratios (from 1:10 to 20:1) based on different killing capacity. For the J7, target cells ($1 \times 10^5$ cells/mL) and effector cells were incubated for 4 h at a ratio from 5:1 based on the cytotoxic ability and cells harvesting. Subsequently, 50 μl samples of culture supernatant were loaded into a 96-well plate with 50 μl of LDH substrate mixture, and incubated in the dark for 30 min at room temperature (detection for cell lysis). Subsequently, 50 μl of stop solution was added to each sample to stop the reaction, and the absorbance was recorded at 490 nm.

## Blocking assay

Antibody-mediated blocking of K562 cell death was performed. Solitary and synergistic use of anti- DNAM-1(BD Bioscience) and anti-NKG2D (R&D Systems) antibodies were explored to evaluate whether they could decrease the cytolytic capacity of CIK cells towards K562 cells. The target cells were incubated with blocking antibodies at a concentration of 20μg/ml for 20 minutes before co-culture with effector cells. K562-only control samples were incubated with isotype control antibodies (BD Bioscience, R&D Systems). These samples were then co-cultured with the effector cells for 4 hours and evaluated by the Annexin V apoptosis kit (BD Bioscience) assisted by flow-cytometry analysis.

## In vivo cytotoxicity in animal experiments

NOD/SCID mice were purchased from the National Laboratory Animal Center and were bred in SPF conditions. Mice at the 6th week of age were used for experiment. In total, $1.33 \times 10^6$ human J7 cells were diluted in 100 μl HBSS and injected into the flank's subcutaneous tissues of these mice. When the tumor grew up to at least 0.2 cm, mice were injected either with

human CIK cells ($1 \times 10^7$ in 400 μl PBS through the tail vein) or with 400 μl PBS every 4 days (N = 5 each). Seven cycles of injection were performed at most. The sizes of the tumors were measured at each injection. After 4 weeks, mice were sacrificed. The tumors infiltrating lymphocytes were analyzed for the presence of human CIK cells, and paraffin embedded specimens were used to detect the presence of human CD4 and CD8 T cells. All experiments were approved by the Animal Care and Use Committee of the Chang Gung Memorial Hospital, Linkou.

## Statistical analysis

Continuous variables are expressed as median [interquartile range (IQR)]. The nonparametric Mann–Whitney U-test was used to compare continuous variables between unmatched groups. When more than 20% of data presented an expected frequency < 5, the Fisher's exact test was substituted for the Chi-square test. The Wilcoxon signed-rank t-test was used when two related samples, matched samples, or repeated measurements on a single sample were compared to assess whether their population means ranked differently. Calculations were made using GraphPad Prism (version 5.0, GraphPad Software, Inc). p-values < 0.05 were considered statistically significant.

## Results

### Phenotypical characteristics of human CIK

CIK cells generated from freshly isolated PBMCs of 10 HCC patients were successfully expanded ex-vivo for 14 days per standard protocol with additions of IFN-γ, anti-CD3, and IL-2. The CIK cells from 5 patients were studied for the immunophenotype and the ex-vivo killing ability. The CIK cells from other 5 patients were tested for in vivo mice model.

The median expansion of CIK cells was 58-fold (range: 20- to 83-fold). The percentage of CD3+ cells increased significantly after expansion (CIK cells vs. PBMCs: 98% vs. 70%, p < 0.05) (Fig 2A). The percentage of CD3+CD8+ cells increased as well after expansion (CIK cells vs. PBMCs: 72% vs. 21%, p < 0.05) (Fig 2B). However, the percentage of CD3+CD4+ cells decreased after expansion (CIK cells vs. PBMCs: 22% vs. 73%, p < 0.05). CD25$^{high}$FoxP3+ T cells was only 1.3% in CIK cells (S1 Fig). The subset of lymphocytes with co-expressions of CD3 and CD56 molecules (CD3+CD56+) was detected at a rate of 15% (range: 9%–25%) in CIK cells, which was significantly higher than that in PBMCs (2.1%, p < 0.05) (Fig 2C), thus indicating approximate 7-fold increment of CD3+CD56+ cells after ex-vivo expansion. In CIK cells, CD56 was major expressed on CD3+CD8+ T cells. Among CD3+CD56+ T cells, CD8+CD4- T cells were 74.3%, but CD8-CD4+ T cells was only 4% (S2 Fig). Conversely, CD3-CD56+ cells (NK cells) were decreased and were negligible in CIK cells (1.1%) compared with PBMCs (7.4%), thus indicating that the NK cell only minimally contributed to the functions of these CIKs.

### NK, inhibitory, and chemokine receptors on human CIK cells

The expression of NK receptors, including NKG2D, DNAM-1, and Nkp30 receptors on CD3+CD8+ cells, was significantly higher in CIK cells, compared with PBMCs (CIK cells vs. PBMCs; NKG2D: 73% vs. 65%, p < 0.05; DNAM-1: 91% vs. 71%, p < 0.05) but similar in NKp30 expressions (CIK vs. PBMCs; 26% vs. 15%, p > 0.05) (Fig 3A). Expression of other NK receptors including NKG2C, NKG2A, KIR, NKp46 and NKp44 were all low in both CIK cells and PBMCs (S3 Fig). The levels of expression of immune checkpoint molecules, including PD-1, CTLA-4, and LAG-3 on CD3+CD8+ and CD3+CD4+ were low (less than 10%) in CIK

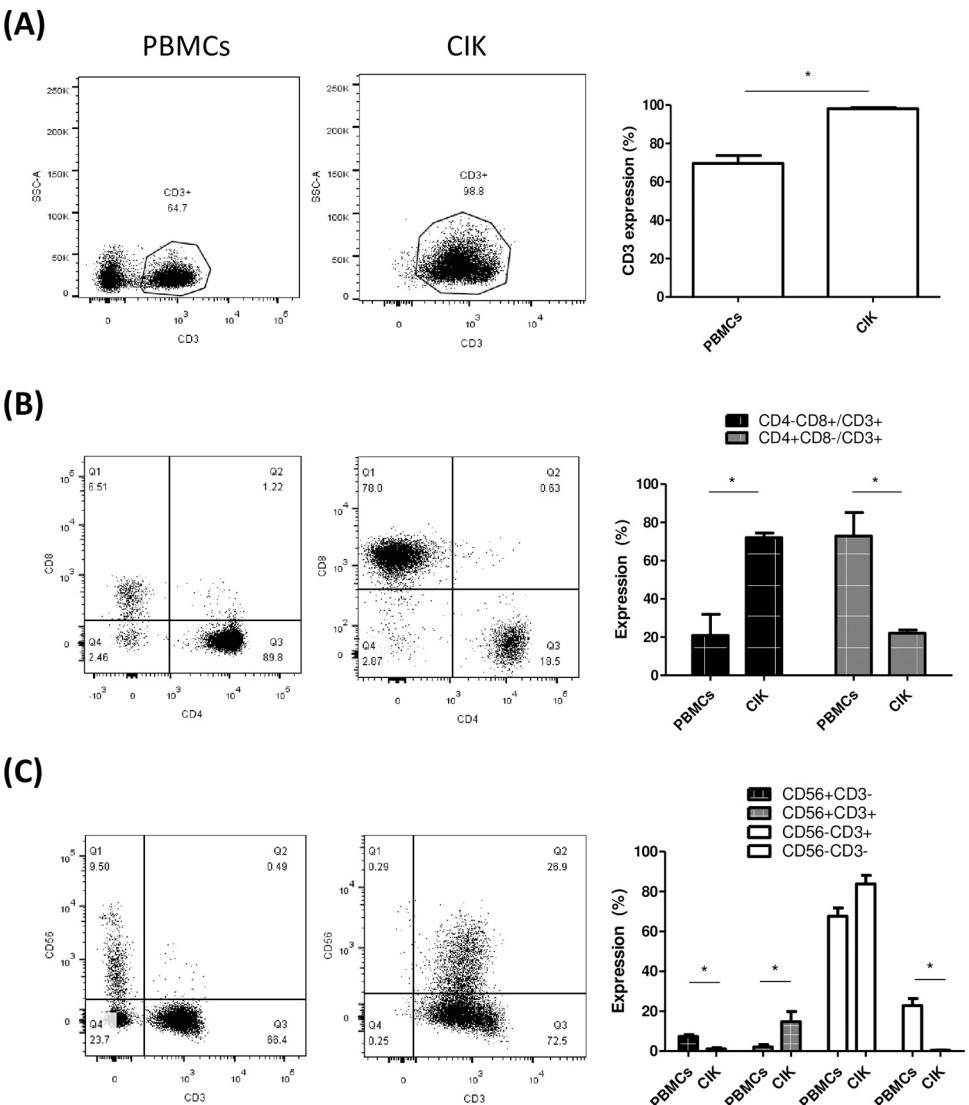

**Fig 2. Phenotypical study of human CIK cells.** A. The dot figures show the gating strategy used for the evaluation of the percentage of CD3+ T cells from a representative subject on preculture PBMCs and the end product of CIK cells. The bar table demonstrates the percentages of CD3+ T cells in PBMCs and CIK cells. B. The dot figures show the gating strategy for the evaluation of the percentages of CD4+ and CD8+ T cells in the total number of T cells from a representative subject on preculture PBMCs and the end product of CIK cells. The bar table demonstrates the percentages of CD4+ and CD8+T cells of PBMCs and CIK cells. C. The dot figures show the gating strategy used for the evaluation of the percentages of CD56 in the total population of CD3+ T cells from a representative subject on preculture PBMCs and the end product of CIK cells. The bar table demonstrates the percentage of CD56+T cells on CD3+ T cells of PBMCs and CIK, respectively (N = 5, *p < 0.05).

cells (Fig 3B); specifically, LAG-3 decreased in CIK cells (7%) compared with PBMCs (36%). Among the chemokine receptors, CD62L and CXCR3 were enhanced in CD3+CD8+ T cells (69% and 84%, respectively) and CD3+CD4+ T cells (74% and 75%, respectively) of CIK cells, representing the trafficking potential to inflamed sites (Fig 3C). The expressions of CCR3, CCR4, CCR5, and CCR7 on CD3+CD8+ cells of CIK cells were all extremely low (0.7%–3.3%).

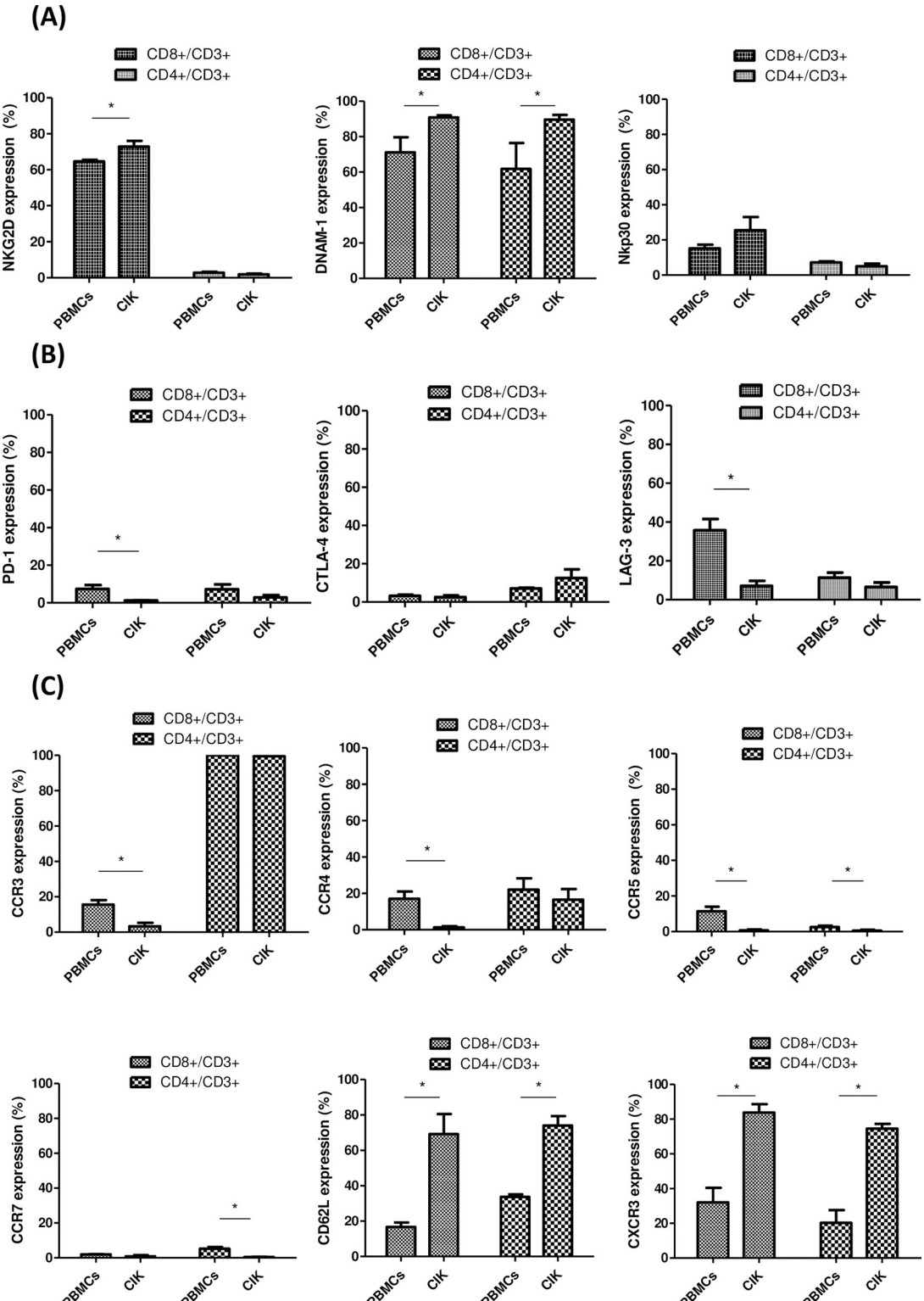

**Fig 3. Expression profile of natural killer (NK) receptors, immune checkpoint molecules, and trafficking/activating molecules on CIK cells.** A. Percentage of NK receptors expression: NKG2D, DNAM-1, and NKp30 on CD8+ T cells (coarse granule bar) and CD4+ T cells (fine granule bar) compared with the total number of CD3+ T cells. B. Percentage of immune checkpoint molecules expressions of PD-1, CTLA-4, and LAG-3 on CD8+ T cells (coarse granule bar) and CD4+ T cells (fine granule bar) with respect to the total number of CD3+ T cells. C. Percentage of chemokine receptors expressions: of CCR3,

CCR4, CCR5, CCR7, CD62L, and CXCR3 on CD8+ T cells (coarse granule bar) and CD4+ T cells (fine granule bar) compared with the total number of CD3+ T cells. (N = 5, *p < 0.05).

### Cytotoxicity of human CIK cells

The ex-vivo cytotoxicity of CIK cells was evaluated using different human cancer cell lines, including the human myelogenous leukemia cell line K562, hepatoma cell line HepG2, and J7 as the target cell lines. As shown in Fig 4A, these CIK possessed cytotoxicity properties to the MHC-deficiency, NK-sensitive K562, and MHC expression HepG2 in terms of induced cell death assessed based on the expressions of Annexin 5 and AAD7 in a dose-dependent E:T ratio manner. By contrast, the cytotoxicity of J7 ATP methods used for assessments yielded a modest cytotoxicity ability at 5:1 E:T ratio compared with positive controls. Furthermore, we found that synergic use of anti-DNAM-1 and anti-NKG2D antibodies decreased the cytolytic ability of CIK cells obtained from HCC patients, suggesting that CIK cells may display cytotoxicity in a DNAM-1 and NKG2D-dependent manner (S4 Fig).

We also evaluated the in vivo antitumor ability, wherein human CIK cells were intravenously infused into NOD/SCID mice (N = 5), which were subcutaneously implanted with the J7 HCC cell line (Fig 4B). In the control group, the tumor grew rapidly from a volume of 113 mm$^3$ to 969 mm$^3$ after it was treated with seven wash cycles with PBS. However, the tumor in the CIK group grew slowly from 59 mm$^3$ to 149 mm$^3$ after 7 cycles of CIK cells (Fig 4B). By contrast, the adoptive transfer of human CIK cells every 4 days suppressed the J7 tumor growth on NOD/SCID mice compared with the control (PBS) group (p < 0.05, Fig 4B). In addition, human immune cells could be detected in the peripheral blood (1.6% of murine cells) and in the tumor microenvironment (15.8% of mouse CD45+CD3+ cells) of NOD/SCID mice after CIK treatment, as evaluated by flow cytometry (Fig 4C). Furthermore, human CD8 and CD4 cells could be detected in the tumor microenvironment by immunohistochemical staining (Fig 4C). All these results demonstrated that these CIK cells execute cytotoxicity against human tumor cells in vitro, and also migrate to the tumor microenvironment to inhibit the growth of the tumors in vivo.

## Discussion

This study focused on the assessment of the immunophenotype and cytotoxicity ability of CIK cells from patients with HCC. The CIK preparation protocol we adopted produced a significant number of cells up to the 14$^{th}$ day of culture. We characterized a deeper effector functional profile of CIK cells from patients with HCC by exploring additional receptor molecules potentially involved in the interaction with tumor targets and the microenvironment. These cells could express high levels of activated NK receptors, including NKG2D and DNAM1. Concerning the immune checkpoint receptors that would dampen the cytotoxic immune response, CIK cells expressed low levels of CTLA-4, PD-1, and LAG-3. Among the chemokine receptors, CD62L and CXCR3 were enhanced in the CD8+ T cells of CIK, thus justifying the trafficking potential to inflamed sites. We also demonstrated the ex-vivo and in vivo cytotoxic abilities of CIK to suppress tumor cells. In addition, human immune cells could be detected in the peripheral blood and on the tumors of NOD/SCID mice after CIK treatment. This is rare in the literature and provides animal model data prior to the initiation of the human CIK study or clinical use [28]. The preclinical data support the potential of adoptive CIK therapy for HCC patients.

Anti-CD3 antibody is a mitogen on T lymphocytes that stimulates T-cell proliferation that is sustained by IL-2 [29]. With this strategy, CIK can be easily expanded multifold (even hundred fold), but with great variety among patients [9]. IFN-γ activates monocytes, then excretes

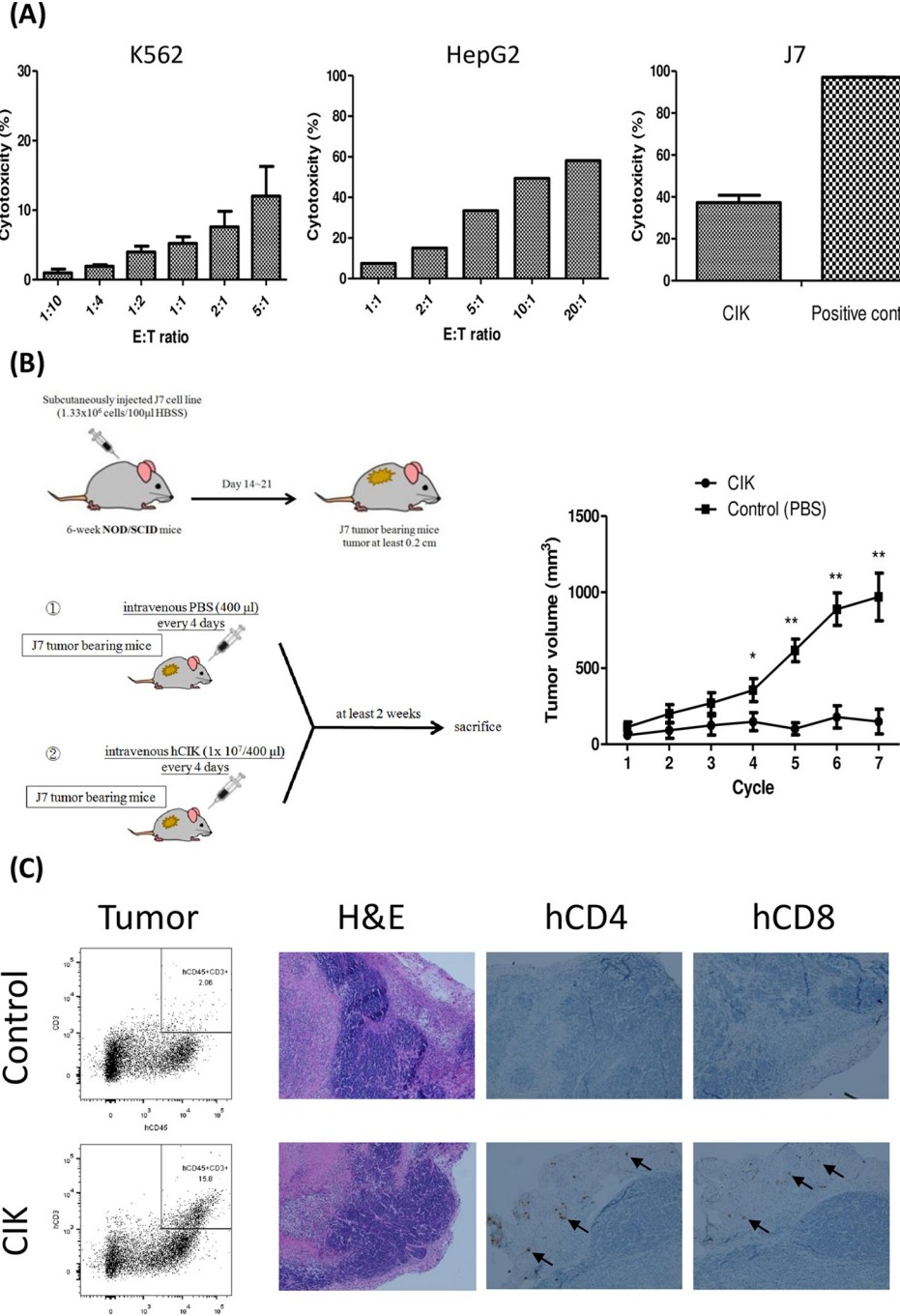

**Fig 4. Ex-vivo and in vivo cytotoxic assay for human CIK cells.** A. Ex-vivo cytotoxicity assay of CIK cells against human cancer cell lines in terms of the percentage of cells expressing Annexin 5 and AAD7. In the case of the K562 cell line, the effector:target ratio (E:T) ranged from 1:10 to 5:1 (N = 3). In the case of the HepG2 cell lines, the E:T ratio ranged from 1:1 to 20:1; (N = 1). In the case of the J7 cell line, we used ATP methods at a fixed E:T ratio at 5:1, the positive control used Triton-X 100 (N = 5). B. The figure on the right illustrates the scheme of human CIK cells adoptive transfer model. Six-week-old NOD/SCID mice were injected with the human J7 cell line. When tumors grew to 0.2 cm, mice were adoptively transferred with human CIK cells every 4 days. The table lists the growth rates of J7 tumors cells on human CIK cells or in PB injected mice (N = 5, *p < 0.05, **p < 0.005). C. The dot figure shows the percentages of human CD45+ cells on tumor infiltrating immune cells on control and CIK cells injected in NOD/SCID mice. The photographs show hematoxylin & eosin staining results from control and human CIK cell-injected mice, as well as the human CD4 and CD8 immunohistochemistry stains on tumor specimens.

soluble proliferating factors or conducts contact-dependent signals to boost T-cell proliferation [30–32]. The variation might have a negative impact on clinical use especially for patients with low-expansion rates [33]. Our data indicated that there were still multifold variations from patients with HCC. However, most PBMCs from patients could be easily expanded at small scales within a laboratory facility. We anticipate that for future clinical level expansion with Good Manufacture Product (GMP) facilities, most HCC patients could still provide a reasonable amount of CIK cells for treatment.

CIK cells were expanded from PBMCs by a nonspecific protocol to enrich CD3 positive T-cell proliferation and activation. This indicates that the CIK cells are a pool of T cells with naive and memory subpopulations without antigen specificity. Previous studies have suggested that the killing mechanism of CIK is nonrestricted MHC [11]. This is particular important as there is increasing evidence that the presence of bystander T cells in the tumor microenvironment may play an important role in mediating cellular immune responses against tumor [34–36]. The concept of cooperation with the adoptive immune cells (whether specific or bystander cells) with endogenous antigen-specific cytotoxic cells that turn the tumor microenvironment to a controlled tumor growth microenvironment is probably more realistic, particular in the era of immune checkpoint blockade therapy [37]. Many have suggested that the bystander killing of CD8+ T cells is driven by the cytokine stimulation in conjunction with NK surface receptor-mediated killing. The K562 cell line does not express the MHC complex, thus further supporting the possible killing mechanism via NK receptors [38]. In the past, the CIK population was categorized by a certain CD56 percentage of expression, a hall marker for NK cells [39], however, our data suggest and are in agreement with others that the presence of NKG2D or DNAM-1 is probably more relevant to the killing ability [13, 40]. The characterization of the expression profile of NK receptors should be essential for future clinical applications other than those related to CD56.

As T-cell exhaustion is a hall marker for tumor immune suppression after protracted T-cell activation. In addition, upregulation of immune checkpoint molecules indicates T-cell exhaustion, and particularly PD-1 expression [41]. Even though with anti-CD3 activation and IL-2 expansion, the CIK cells from our laboratory express very low levels of PD-1, CTLA4, and Tim-3, outcomes are compatible with literature publications. The significantly increased expression of LAG3 is uncertain in CIK, while the role of LAG-3 on CD8+ T cells may comprise suppressive functions [42]. However, whether anti-LAG3 Ab could potentiate the effect of CIK deserves additional studies. Our data suggest that CIK cells do not possess an exhausted phenotype that could potentially maintain their activation in vivo. This was further confirmed in our in vivo animal study which confirmed the presence of human CIK cells in both PBMCs and the tumor microenvironment.

CIK cells are from PBMCs and the circulating cells are usually cells with central memory phenotype that are not ready to enter the inflamed site. Our analysis found that CIK cells express higher levels of CD62L and CXCR3, and lower CCR7 levels. The presence of CD62L suggested that many of these cells remain in naïve status, but the presence of CXCR3 indicates that these cells are able to migrate to inflamed sites [43]. Lower CCR7 expressions suggest there is not central memory [44]. In combination with the animal experiment, our data support the fact that CIK cells are inflamed tissue-homing T cells with effector functions.

The role of CIK for HCC was promising. Hui et al. reported that CIK therapy significantly improved disease-free survival, but not the overall survival in a randomized trial [23]. Weng et al. used intrahepatic artery CIK infusions in a randomized study as adjuvant therapy, and demonstrated a benefit on recurrence free survival [45]. While a prospective randomized trial by Yu et al. reported significant improvement in OS and DFS by CIK cells in HCC patients who were not candidates for surgical intervention [46]. A meta-analysis summarized these

data and validated the benefits on DFS for HCC CIK cells after local treatment [47]. However, some clinicians would criticize the methodology, study design, or heterogenous patients that may diminish the value of these results. The Korean study enrolled only stage I and II HCC patients who had already completed local treatments. The results showed that in the 5-year follow up, that there was a 37% death rate reduction in favor of adoptive immunotherapy with CIK on 230 patients. These encouraging results suggested that HCC should be one of the most valuable cancer types for CIK clinical utilities.

The mice model is an important issue for human anti-cancer study. NOD/SCID mice contain impaired T cell, B cell lymphocyte and NK cell function, and lack macrophage and dendritic cell activity as well as reduced complement activity. Therefore, NOD/SCID mice exhibit a sufficient ability to support engraftment with human cancer tissues including liver cancer [48–52], and with hematopoietic cells [53–55]. It provides a reliable model of anti-cancer therapy screening. Although, many cancers still fail to engraft efficiently in NOD/SCID mice, largely due to remaining NK cell activity and other residual innate immune function. This leakiness may be eliminated in the RAG-1 and RAG-2 mice, or especially the NSG (NOD-SCID-IL2γ$^{-/-}$) mice, produced by the targeted mutation of the IL-2 receptor γ-chain locus in a previously bred NOD-SCID strain, lacking T cells, B cells, macrophages, NK and NKT cells, may also be platforms to overcome the human engraft failure. Beyond the endogenous immune cells, xenograft mouse models affect the tumor growth includes the influences on immune cell infiltration and angiogenesis. The immune cell activation patterns or antigen expression between the mouse and the human immune system were other hurdles. Advances in humanized mouse models are remarkable but are not yet fully established in HCC research [48]. Nevertheless, immunosuppressive cells such as myeloid-derived suppressor cells (MDSC), T-regulatory cells and M2 macrophages could also play an important role in the interplay between a patient's tumor and the immune system. One previous study showed human CIK cells could be affected by MDSC [56]. Taken together, there are our limitations to establish an artificial environment to define the exact anticancer effect of human CIK cells.

The limitation of our study was that we did not conduct this protocol at a large-scale level and used the GMP facility. This pivotal study is a proof-of-concept, and the proof of feasibility aims to define the immunophenotype of CIK cells and validate their cytotoxic properties. The large-scale, GMP facility production protocol uses a stricter procedure, high-quality medium, and cytokines for human use purposes. The culture density will be much lower than the current laboratory that allows an increased expansion capacity for these cells. Therefore, we expect that our next preclinical preparation would produce folds of CIK cells with identical immune phenotypes. Additionally, there is still a concern on whether the patient characteristics will affect the quality and quantity of CIK cells given the observed variation in the CIK cell population. We found a variation in the expansion of fold changes among different patients. However, the number of patients was still limited to allow us to conclude any potential factors associated with the variation. For the immunophenotype, it seems that the variation was much lower, thus suggesting that the quality of CIK cells could be maintained among different individuals. We only enrolled HCC patients. Whether the CIK population is limited to HCC patients remains elusive, despite prior report findings that supported the fact that the CIK treatment series is not limited to HCC. However, we still need to enroll patients affected by other cancer types to validate the applicability of this protocol to other cancer types. By using a mathematical approach, one study suggested that calculation of CD8+ or CD4+ T-cell percentages on early days could predict the CIK expansion quality [57]. Other strategies were used to improve the expansion and function of CIK cells, such as the addition or replacement of cytokines, like IL-7, IL-15 or thymoglobulin, instead of IL-2 [58–61], addition of allogeneic APCs, depletion of regulatory T cells, or pharmacological interventions [15, 62].

## Conclusions

In summary, we characterized the immunophenotype of CIK cells and proved its cytotoxic ability against human cell lines. We set up the quality control parameters for the CIK cells for future clinical applications other than the classical CD56 surface marker. This study further supports the possible mechanism of the killing effects of CIK cells as a group of highly activated, inflamed site-trafficking bystander T cells. The cooperated potential with antigen-specific T cells highlights an important role of adoptive CIK therapy for the future clinical application for HCC.

## Supporting information

**S1 Fig. CD25<sup>high</sup>FoxP3+ T cells on CIK cells.** Expression of CD25<sup>high</sup>FoxP3+ T cells were low on CIK cells and PBMCs (N = 3).
(TIF)

**S2 Fig. CD3+CD56+ T cells on CIK cells.** CD56 was major expressed on CD3+CD8+ T cells in CIK cells (N = 5).
(TIF)

**S3 Fig. Expression profile of partial NK receptors on CIK cells.** Expression of NKG2C, NKG2A, KIR, NKp46 and NKp44 were low on CIK cells and PBMCs (N = 5).
(TIF)

**S4 Fig. Bocking assay of CIK cells by anti-DNAM-1 and anti-NKG2D.** The blocking assay was performed to explore the potential mechanism for the cytolytic activity of CIK cells from HCC patients via a DNAM-1 and NKG2D-dependent manner (N = 2). The synergic use of anti-DNAM-1 and anti-NKG2D antibodies decreased the cytolytic ability of CIK cells obtained from HCC patients.
(TIF)

## Acknowledgments

We thank Cell Therapy Center in Chang Gung Memorial Hospital at Taoyuan for technical support. We also like to thank Hsin-Yi Lai for assistance in performing experiments.

## Author Contributions

**Conceptualization:** Chan-Keng Yang, Tse-Ching Chen, Yung-Chang Lin, Chun-Yen Lin.

**Data curation:** Chan-Keng Yang, Chien-Hao Huang, Ching-Hsun Hu, Jian-He Fang, Yung-Chang Lin.

**Formal analysis:** Chan-Keng Yang, Ching-Hsun Hu, Jian-He Fang.

**Funding acquisition:** Chan-Keng Yang, Yung-Chang Lin, Chun-Yen Lin.

**Investigation:** Chan-Keng Yang, Yung-Chang Lin, Chun-Yen Lin.

**Methodology:** Chan-Keng Yang, Chien-Hao Huang, Ching-Hsun Hu, Jian-He Fang, Tse-Ching Chen, Yung-Chang Lin.

**Project administration:** Chan-Keng Yang, Yung-Chang Lin, Chun-Yen Lin.

**Resources:** Chan-Keng Yang, Chien-Hao Huang, Yung-Chang Lin, Chun-Yen Lin.

**Supervision:** Tse-Ching Chen, Yung-Chang Lin, Chun-Yen Lin.

**Validation:** Chan-Keng Yang, Ching-Hsun Hu, Jian-He Fang, Yung-Chang Lin.

**Visualization:** Chien-Hao Huang, Jian-He Fang, Tse-Ching Chen, Yung-Chang Lin, Chun-Yen Lin.

**Writing – original draft:** Chan-Keng Yang, Yung-Chang Lin.

**Writing – review & editing:** Chan-Keng Yang, Yung-Chang Lin, Chun-Yen Lin.

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
