## [Decision Letter · Decision Letter 0]

2 Sep 2022

PONE-D-22-17512Immunophenotype and antitumor activity of cytokine-induced killer cells from patients with hepatocellular carcinomaPLOS ONE

Dear Dr. Lin,

Thank you for submitting your manuscript to PLOS ONE. After careful consideration, we feel that it has merit but does not fully meet PLOS ONE’s publication criteria as it currently stands. Therefore, we invite you to submit a revised version of the manuscript that addresses the points raised during the review process.

The manuscript has been evaluated by one reviewer, and his comments are available below.<o:p></o:p>

The reviewer has raised a number of concerns that need attention. He requests additional information on methodological aspects of the study such as the definition of CIK and the number of patients used. <o:p></o:p>

Could you please revise the manuscript to carefully address the concerns raised?<o:p></o:p>

Please note that we have only been able to secure a single reviewer to assess your manuscript. We are issuing a decision on your manuscript at this point to prevent further delays in the evaluation of your manuscript. Please be aware that the editor who handles your revised manuscript might find it necessary to invite additional reviewers to assess this work once the revised manuscript is submitted. However, we will aim to proceed on the basis of this single review if possible. <o:p></o:p>

We look forward to receiving your revised manuscript.

Kind regards,

Lorena Verduci

Staff Editor

PLOS ONE

Journal Requirements:

"This work was supported by CMRPG3F1301, CMRPG3J0871, CMRPG5J0101 and CMRPG5K0221 from Medical Research Project Fund, Chang Gung Memorial Hospital, Taiwan and MOST 107-2314-B-182A-157 from Ministry of Science and Technology, Taiwan."

"This work was supported by CMRPG3F1301, CMRPG3J0871, CMRPG5J0101 and CMRPG5K0221 from Medical Research Project Fund, Chang Gung Memorial Hospital, Taiwan and MOST 107-2314-B-182A-157 from Ministry of Science and Technology, Taiwan."

7. Please upload a new copy of Figure 4 as the detail is not clear. Please follow the link for more information: https://blogs.plos.org/plos/2019/06/looking-good-tips-for-creating-your-plos-figures-graphics/"" https://blogs.plos.org/plos/2019/06/looking-good-tips-for-creating-your-plos-figures-graphics/

Reviewers' comments:

Reviewer's Responses to Questions

**Comments to the Author**

1. Is the manuscript technically sound, and do the data support the conclusions?

Reviewer #1: Partly

2. Has the statistical analysis been performed appropriately and rigorously? 

Reviewer #1: Yes

3. Have the authors made all data underlying the findings in their manuscript fully available?

Reviewer #1: Yes

4. Is the manuscript presented in an intelligible fashion and written in standard English?

Reviewer #1: Yes

5. Review Comments to the Author

Reviewer #1: Major Points:

Please elaborate what is new finding in this paper?

Could you correlate expression of DNAM or NKG2D on K562 and HepG2 with the susceptibility to killing by CIK?

CIK from how many HCC patients were tested?

How are you defining CIK here; CD3+CD4+ T cells, CD3+CD8+ T cells or both?

Is CD56 expressed on CD3+CD4+ T cells or only on CD3+CD8+ T cells?

What was the %age of CD25highFoxP3+ T cells in expanded cells?

What would be the effects on CIK, if IL-15 is added to expand and induce CIK?

Why did not you investigate expression of other NK cell receptors like NKG2C, NKG2A, KIR, NKp46, etc?

Minor points:

Page 11: The results of other treatments for palliative purpose remain……(not remains).

Page 13: A renowned worldwide cancer treatment guideline has adopted………(provide ref).

6. PLOS authors have the option to publish the peer review history of their article (what does this mean?). If published, this will include your full peer review and any attached files.

Reviewer #1: No

---

## [Author Response · Author response to Decision Letter 0]

14 Oct 2022

Dear Editor and Reviewers, 

Please find enclosed a revised manuscript (PONE-S-22-22620) from my group, which we would like to have considered for publication in the PLoS ONE: “Immunophenotype and antitumor activity of cytokine-induced killer (CIK) cells from patients with hepatocellular carcinoma” by Yang et al. 

In this revised manuscript, we have addressed in full the comments of the Reviewer and Journal Requirements, performed the additional data analyses and experiments as requested and modified the manuscript accordingly. 

In this study, we report an in-depth description of convention CIK preparation, defining its phenotypical features, functional assessment and in vitro and in vivo cytotoxic studies. Other than previous literature, we emphasized the importance of NK surface receptors, chemokine receptors in defining the CIK population and function. We demonstrated the intravenous human CIK cells could infiltrate to tumor site and contain anti-tumor ability in xenograft mouse model. This finding had not proved in other papers. This study further supports the possible mechanism of the killing effects of CIK cells as a group of highly activated, inflamed site-trafficking bystander T cells. Our study adds more relevant evidence to the current understanding of CIK and provides sound rationale for the CIK therapy.

We have included in this submission a “List of amendments to the manuscript”. Furthermore, all the modifications to the manuscript are detailed in the “Point-by-point response to reviewer’s comments”. Changes made in the manuscript were also tracked for an expedient review; both tracked and untracked versions are submitted. We hope you will agree that the data and conclusions here will be of interest to the wide readership of PLoS ONE and look forward to hearing your favorable response in the near future.

Sincerely,

Yung-Chang Lin, M.D.

---

## [Decision Letter · Decision Letter 1]

21 Nov 2022

PONE-D-22-17512R1Immunophenotype and antitumor activity of cytokine-induced killer cells from patients with hepatocellular carcinomaPLOS ONE

Dear Dr. Chang Lin,

Thank you for submitting your manuscript to PLOS ONE. After careful consideration, we feel that it has merit but does not fully meet PLOS ONE’s publication criteria as it currently stands. Therefore, we invite you to submit a revised version of the manuscript that addresses the points raised during the review process. Kindly address comments raised by reviewer 2 and resubmit.

We look forward to receiving your revised manuscript.

Kind regards,

Afsheen Raza, PhD

Academic Editor

PLOS ONE

Additional Editor Comments:

Kindly address the comments by reviewer 2 for manuscript revision

Reviewers' comments:

Reviewer's Responses to Questions

**Comments to the Author**

1. If the authors have adequately addressed your comments raised in a previous round of review and you feel that this manuscript is now acceptable for publication, you may indicate that here to bypass the “Comments to the Author” section, enter your conflict of interest statement in the “Confidential to Editor” section, and submit your "Accept" recommendation.

Reviewer #1: All comments have been addressed

Reviewer #2: (No Response)

2. Is the manuscript technically sound, and do the data support the conclusions?

Reviewer #1: (No Response)

Reviewer #2: (No Response)

3. Has the statistical analysis been performed appropriately and rigorously? 

Reviewer #1: (No Response)

Reviewer #2: (No Response)

4. Have the authors made all data underlying the findings in their manuscript fully available?

Reviewer #1: (No Response)

Reviewer #2: (No Response)

5. Is the manuscript presented in an intelligible fashion and written in standard English?

Reviewer #1: (No Response)

Reviewer #2: (No Response)

6. Review Comments to the Author

Reviewer #1: (No Response)

Reviewer #2: Manuscript ID: PONE-D-22-17512R1

Manuscript Title: Immunophenotype and antitumor activity of cytokine-induced killer cells from patients with hepatocellular carcinoma

The authors activated human cytokine-induced killer (CIK) cells ex-vivo and expanded from PBMCs from hepatocellular carcinoma (HCC) patients. They evaluated immunophenotyping and the ex-vivo killing ability of human CIK cells. Furthermore, they intravenously injected human CIK cells into NOD/SCID mice to evaluate the in vivo anticancer ability. They used human HCC cell line, J7, bearing NOD/SCID mice model in this study. This is an important subject and well written manuscript but there are several points should be addressed before its publication.

Non-obese-diabetic–severe combined immunodeficient mice (NOD–SCID) which were reported to lack innate immunity but are limited owing to a residual appearance of T and B cells and remaining natural killer (NK) cells (Cancer Res. 76, 6153–6158 (2016)). Recombination activating gene (RAG)-knockout mice overcome the residual appearance of adaptive immune cells but still show NK cells (Cancer Res. 76, 6153–6158 (2016)). NSG (NOD/SCID-IL2g-/−) strain mice, produced by the targeted mutation of the IL-2 receptor g-chain locus in a previously bred NOD–SCID strain, lack T cells, B cells, macrophages, NK cells and natural killer T cells (NKT) and have become an increasingly used platform for xenograft models (Cancer Res. 76, 6153–6158 (2016); Cancer Lett. 344, 13–19 (2014)). Therefore, we cannot tell the origin of antitumor effect in this study. The authors are better to use RAG-knockout mice or NSG (NOD/SCID-IL2g-/−) mice model in this study.

Another point to be addressed here is that human CIK cells could be affected by immunosuppressive cells (J Hepatol. 2019 Mar;70(3):449-457). Therefore, we need human immunosuppressive cells in murine HCC models to tell the exact anticancer effect of human CIK cells in vivo system.

7. PLOS authors have the option to publish the peer review history of their article (what does this mean?). If published, this will include your full peer review and any attached files.

Reviewer #1: No

Reviewer #2: No

---

## [Author Response · Author response to Decision Letter 1]

9 Dec 2022

Thank you very much for these essential recommendations. The mice model is really an important issue for human anti-cancer study. We used the NOD/SCID mice in this study due to the previous well established engraftment mice model for liver cancer [1-5]. In our control group subcutaneously implanted with the J7 HCC cell line (N=5), the tumor grew well and rapidly (Fig. 4B); however, the tumor in the CIK group grew slowly, suggesting that human CIK could help inhibit human tumor growth in immune compromised mice. We think NOD/SCID mice model is reasonable for this study. Indeed, it’s one of our limitations to establish human immunosuppressive tumor environment to tell the exact anticancer effect of CIK cells, especially that human CIK cells could be affected by MDSC. 

 We added a new paragraph in “Discussion” to describe the reason of our NOD-SCID mice model and the limitation of lack of human immunosuppressive cells in our model. We hope you can accept our explanations and revision. Thank you again for your kindly consideration.

---

## [Editor Report · Decision Letter 2]

21 Dec 2022

Immunophenotype and antitumor activity of cytokine-induced killer cells from patients with hepatocellular carcinoma

PONE-D-22-17512R2

Dear Dr. Lin,

We’re pleased to inform you that your manuscript has been judged scientifically suitable for publication and will be formally accepted for publication once it meets all outstanding technical requirements.

Kind regards,

Afsheen Raza, PhD

Academic Editor

PLOS ONE
---

## [Editor Report · Acceptance letter]

26 Dec 2022

PONE-D-22-17512R2 

Immunophenotype and antitumor activity of cytokine-induced killer cells from patients with hepatocellular carcinoma 

Dear Dr. Lin:

I'm pleased to inform you that your manuscript has been deemed suitable for publication in PLOS ONE. Congratulations! Your manuscript is now with our production department. 

Kind regards, 

on behalf of

Dr. Afsheen Raza 

Academic Editor

PLOS ONE